# Effector-Memory B-Lymphocytes and Follicular Helper T-Lymphocytes as Central Players in the Immune Response in Vaccinated and Nonvaccinated Populations against SARS-CoV-2

**DOI:** 10.3390/vaccines10101761

**Published:** 2022-10-20

**Authors:** Lorenzo Islas-Vazquez, Marisa Cruz-Aguilar, Henry Velazquez-Soto, Aida Jiménez-Corona, Sonia Mayra Pérez-Tapia, Maria C. Jimenez-Martinez

**Affiliations:** 1Department of Immunology and Research Unit, Institute of Ophthalmology “Conde de Valenciana Foundation”, Mexico City 06800, Mexico; 2Department of Ocular Epidemiology, Institute of Ophthalmology “Conde de Valenciana Foundation”, Mexico City 06800, Mexico; 3Health Secretariat, General Directorate of Epidemiology, Mexico City 01480, Mexico; 4Unidad de Desarrollo e Investigación en Bioterapéuticos (UDIBI), Escuela Nacional de Ciencias Biológicas, Instituto Politécnico Nacional, Mexico City 11340, Mexico or; 5Laboratorio Nacional para Servicios Especializados de Investigación, Desarrollo e Innovación (I+D+i) para Farmoquímicos y Biotecnológicos, LANSEIDI-FarBiotec-CONACyT, Mexico City 11340, Mexico; 6Departamento de Inmunología, Escuela Nacional de Ciencias Biológicas, Instituto Politécnico Nacional (ENCB-IPN), Mexico City 11340, Mexico; 7Department of Biochemistry, Faculty of Medicine, National Autonomous University of Mexico, Mexico City 04510, Mexico

**Keywords:** SARS-CoV-2, vaccine, COVID-19, B-lymphocytes, follicular helper T-lymphocytes (TFH)

## Abstract

Vaccines have been recognized as having a central role in controlling the COVID-19 pandemic; however, most vaccine development research is focused on IgG-induced antibodies. Here, we analyzed the generation of IgGs related to SARS-CoV-2 and the changes in B- and T-lymphocyte proportions following vaccination against COVID-19. We included samples from 69 volunteers inoculated with the Pfizer-BioNTech (BNT162b2), Astra Zeneca (AZD1222 Covishield), or Sputnik V (Gam-COVID-Vac) vaccines. IgGs related to SARS-CoV-2 increased after the first vaccine dose compared with the nonvaccinated group (Pfizer, *p* = 0.0001; Astra Zeneca, *p* < 0.0001; Sputnik V, *p* = 0.0089). The results of the flow cytometry analysis of B- and T-lymphocytes showed a higher proportion of effector-memory B-lymphocytes in both first and second doses when compared with the nonvaccinated subjects. FcRL4+ cells were increased in second-dose-vaccinated COVID-19(−) and recovered COVID-19(+) participants when compared with the nonvaccinated participants. COVID-19(−) participants showed a lower proportion of follicular helper T-lymphocytes (TFH) in the second dose when compared with the first-vaccine-dose and nonvaccinated subjects. In conclusion, after the first vaccine dose, immunization against SARS-CoV-2 induces IgG production, and this could be mediated by TFH and effector-memory B-lymphocytes. Our data can be used in the design of vaccine schedules to evaluate immuno-bridging from a cellular point of view.

## 1. Introduction

The coronavirus disease 2019 (COVID-19) pandemic is caused by severe acute respiratory syndrome coronavirus 2 (SARS-CoV-2), a newly emerged coronavirus [1]. To date, SARS-CoV-2 has caused over 200 million cases and around 5 million deaths worldwide. Vaccines represent the most efficient means to control the COVID-19 pandemic. Since the beginning of the pandemic, about 200 vaccines have entered the biopharmaceutical development process. The vaccination aim is to achieve a robust immune response against virus structural proteins (S, M, or N proteins) [2,3,4] similar to that seen in convalescent individuals [5,6]. In Mexico, there are a variety of vaccine platforms authorized for emergency use against COVID-19, such as Pfizer-BioNTech, Moderna, Astra Zeneca, CanSino Biologic, Sputnik V, Sinovac, Sinopharm, Covaxin, and Jansen from Johnson & Johnson. For several of these vaccines, the immune response has not been characterized, but sufficient clinical evidence related to their efficacy has been reported for approval by the local regulatory authority.

Among vaccines that currently have sufficient evidence of their efficacy are the BNT162b2 (Pfizer-BioNTech) and SPIKEVAX (Moderna, Cambridge, MA, USA) vaccines, whose mRNA platforms induce seroconversion with specific IgG antibodies against the SARS-CoV-2 Spike protein region when determined at 21 and 28 days, respectively [7,8]. The efficacy of the AZD1222 (Astra Zeneca, Cambridge, UK) vaccine, with the nonreplicating chimpanzee adenovirus (ChadOx1) platform, was evaluated by the generation of IgG antibodies against the RBD Spike protein region at 14 and 28 days after the first and second dose of vaccine, respectively [9]. In addition, the production of IFN gamma by Th1 lymphocytes was detected [9]. For the Ad26.COV2-S vaccine, which uses a nonreplicating adenovirus 26 platform, neutralizing antibodies can be detected by serological determination at 29 days [10].

Although the immune response against SARS-CoV-2 has been characterized in the infected population, the development of immunity after administering antiviral COVID-19 vaccines has not yet been sufficiently studied [11,12]. Descriptions of the generation of antibodies against SARS-CoV-2, predominance of these antibodies over time, and changes induced in the cellular immune response in the short and medium term after the vaccination can contribute to efforts to stop the pandemic. Therefore, this study aimed to analyze the generation of IgG related to SARS-CoV-2 and the changes in the proportions of B- and T-lymphocyte subpopulations induced by the vaccine against COVID-19.

## 2. Materials and Methods

### 2.1. Population Studied

A prospective cohort study was carried out including samples from 69 volunteers, of which 34 participants were not yet vaccinated, and 36 had already received the first vaccine dose. We followed up throughout the vaccination schedule, collecting 34 samples before vaccination, 54 samples after the first vaccine dose, and 43 samples after the second vaccine dose. The vaccinated participants were inoculated with the Pfizer-BioNTech (BNT162b2), Astra Zeneca (AZD1222 Covishield), or Sputnik V (Gam-COVID-Vac) vaccines. The interval between the first and second dose was one month for the Pfizer-BioNTech (BNT162b2) and Sputnik V (Gam-COVID-Vac) vaccines and two months for the Astra Zeneca (AZD1222 Covishield) vaccine. In addition, we considered the clinical history of COVID-19. Diagnosis of COVID was performed with clinical data and confirmed by rtPCR; all recovered COVID-19(+) patients coursed a mild-to-moderate illness. In addition, 14 days after the onset of infection, a second rtPCR was performed. The infection recovery occurred between three and four months before the blood sample collection. Patients with negative rtPCR test at this time were included in this study. The demographic characteristics of participants are shown in Table 1. Written informed consent was obtained from all participants before enrolling in the study and peripheral blood collection. The Science (CI-055-2021), Biosecurity (CB-056-2021), and Bioethics (CEI-2021/10/09) Committees from the Institute of Ophthalmology “Conde de Valenciana Foundation” approved this study.

### 2.2. Blood Sample Collection

A total of 10–12 mL of peripheral blood was obtained through venipuncture. Peripheral blood samples were collected before the first vaccine dose and 14 ± 5 days after the first and or second vaccine application (Figure 1). To obtain the serum, one blood sample was allowed to clot for 10 min at room temperature and centrifuged at 3500 rpm (1000× *g*) for 7 min. Then, the serum was aliquoted and stored at −20 °C until the enzyme-linked immunosorbent assay (ELISA) was performed. Another blood sample was immediately used for complete blood count and flow cytometry analysis.

### 2.3. Detection of IgGs Related to SARS-CoV-2 by Enzyme-Linked Immunosorbent Assay (ELISA)

IgGs related to SARS-CoV-2 against the RBD domain of S protein were detected in the serum collected from vaccinated and nonvaccinated participants using an ELISA Kit UDITEST-V2G (IPN, Mexico City, Mexico), according to the manufacturer’s instructions [13]. The optical density (OD) was measured at 450 nm in a Multiskan Ascent spectrophotometer (Thermo Scientific, Waltham, MA, USA). Each sample was run in duplicate, and the values were averaged.

### 2.4. Total Count of Neutrophils, Lymphocytes, and Neutrophil–Lymphocyte Ratio (NLR)

The total number of neutrophils and lymphocytes was obtained from the complete blood count data performed in an Automated Hematology Analyzer XS-1000i (Sysmex, Kobe, Japan). The neutrophil–lymphocyte ratio (NLR) was defined as the proportion of the absolute neutrophil count to absolute lymphocyte count.

### 2.5. Multiparametric Flow Cytometry Analysis

Panel of antibodies for phenotyping of B-lymphocytes and follicular helper T-lymphocytes (TFH). The following labeled anti-human monoclonal antibodies (MoAbs) were used: BV605 anti-CD3 (HIT3a Clone, BD #564712), PECy5 anti-CD4 (RPA-T4 Clone, BD #555348), Alexa Fluor 488 anti-CXCR5 (RF8B2 clone, BD #558112), PE anti-CCR7 (3D12 clone, BD #552176), PECy7 anti-CD45RO (UCHL1 clone, BD #560608), and APC-H7 anti-CD45RA (HI100 clone, BD #560674), FITC anti-CD19 (HIB19 clone, BD #555412), PE anti-CD21 (B-ly4 clone, BD, #555422), APC-H7 anti-CD27 (M-T271 clone, BD, #560222), and Alexa Fluor 647 anti-FcRL4 (A1 clone, BD, #566587) from BD (San Jose, CA, USA).

CD4+ T-lymphocytes and CD19+ B-lymphocytes staining procedure and flow cytometry analysis. Briefly, 100 μL of complete blood was placed in a cytometric tube, and a viability staining for discriminated live or dead cells was performed by employing Ghost Violet 450 (#13-0863-T100, San Diego, CA, USA) from Tonbo bioscience, according to the manufacturer’s instructions. Then, anti-CD19, anti-CD21, anti-CD27, and anti-FcRL4 MoAbs or anti-CD3, anti-CD4, anti-CD45RO, anti-CD45RA, anti-CCR7, and anti-CXCR5 MoAbs were added. The tubes were incubated at room temperature for 30 min. After the incubation, 1 mL of the BD FACS Lysing solution (BD, San José, CA, USA) was added and incubated at room temperature for 12 min, washed, and centrifuged. Finally, the cellular pellet was resuspended and fixed with 300 μL of the BD Stabilizing Fixative solution (San Jose, CA, USA).

Events were acquired with a BD FACSVerse (San Jose, CA, USA) cytometer using BD FACSuite software. First, the cells were gated in an FSC-A vs. Ghost Violet to exclude dead cells; later, in the FSC-A vs. FSC-H dot plot, we excluded doublets. Then, the lymphocyte region was selected using an FSC vs. SSC dot plot; 20,000 events were acquired from this population. Next, an SSC-A vs. CD3 followed by CD3 vs. CD4 or an SSC-A vs. CD19 dot plot was made. From the gated-CD3+CD4+ region, a CXCR5 vs. CCR7 plot was used to identify and quantify the TFH population. Additionally, from the gated-CD4 region, a CD45RO vs. CD45RA plot was used to identify and quantify the memory T-lymphocyte population. Conversely, from the gated-CD19 region, a CD27 vs. CD21 plot was used to identify and quantify the memory and effector-memory B-lymphocyte populations. In addition, from the populations mentioned above, the expression of FcRL4 was evaluated (Figure 2). Positive and negative gates for each molecule were verified on the basis of “fluorescence-minus-one (FMO)” control (Appendix A).

### 2.6. Statistical Analysis

For the parametric distribution, the Shapiro–Wilk test was used. Statistical analysis was performed using the Mann–Whitney test to compare between COVID-19(−) and recovered COVID-19(+) participants in nonvaccinated, vaccinated with the first dose, or vaccinated with the second dose. The Kruskal–Wallis test, followed by Dunn’s multiple comparison test, was performed to compare groups before and throughout the vaccine schedule. The Wilcoxon test for paired samples or the Friedman test with Dunn’s multiple comparisons was used for the longitudinal analysis throughout the follow-up of the vaccination schedule, when the participants had one or two vaccine doses, respectively.

All results are shown as the mean ± standard deviation (SD). Finally, random-effects generalized least-squares regression (GLS regression) was used to evaluate changes on the optical density (OD) after adjustment by TFH, memory B-lymphocytes or effector-memory B-lymphocytes, and COVID-19 background. Analyses were performed using the statistical program Graph Pad Prism 8 (GraphPad Software, La Jolla, CA, USA and Stata 15.1, Stata Corp., College Station, TX, USA). Values of *p* < 0.05 were considered statistically significant.

## 3. Results

### 3.1. Distinct Vaccine Platforms Induced Similar Production of Antibodies against SARS-CoV-2

First, we began by determining the IgG production against SARS-CoV-2 according to the vaccine platform. All analyzed vaccines significantly increased the optical density (OD) after the first dose compared with the nonvaccinated group (Pfizer *p* = 0.0001, Astra Zeneca *p* < 0.0001, and Sputnik V *p* = 0.0089) (Figure 3A and Appendix A). Interestingly, with the first dose, we detected no significant differences in the OD between the distinct vaccine types. Based on these results, for the following analyses, we evaluated nonvaccinated, vaccinated with the first dose, and vaccinated with the second dose independently of the vaccine type. In addition, analysis with age stratification (subjects under 30 years and subjects over 30 years) was carried out, and no differences were detected in the vaccinated samples (Appendix A).

A second analysis was performed considering COVID-19 history before immunization and determining immunization-induced IgG levels; we analyzed the OD in recovered COVID-19(+) and COVID-19(−) participants for each group of nonvaccinated, vaccinated with the first dose, and vaccinated with the second dose. As expected, in the participants with previous COVID-19, we detected an induction of IgGs related to SARS-CoV-2 before the vaccine (OD mean = 0.8). Furthermore, the OD detected in recovered COVID-19(+) participants was significantly greater than in the participants naïve to COVID-19 in samples with the first (*p* < 0.0001) and second vaccine doses (*p* = 0.0083). Interestingly, with the first vaccine dose, we detected a greater increase of the OD in both recovered COVID-19(+) and COVID-19(−) participants compared with the nonvaccinated samples (*p* < 0.0001), whereas with the second vaccine dose, an increase in the OD was detected only in the COVID-19(−) samples compared with the samples with the first dose (*p* < 0.0001) (Figure 3B).

### 3.2. Immunization against SARS-CoV-2 Does Not Induce Differences in Total Neutrophils, Although a Decrease in Lymphocytes Is Detected

Neutrophils are a major component of the leukocyte population capable of releasing large amounts of reactive oxygen species and pro-inflammatory cytokines, and lymphocytes are an important part of the adaptive immune response against the virus. Therefore, we analyzed the total count of neutrophils and lymphocytes in COVID-19(−) and recovered COVID-19(+) participants in three groups: nonvaccinated, vaccinated with the first dose, and vaccinated with the second dose. All neutrophil and lymphocyte values were in the normal range defined by the hematic biometry (Figure 4A,B). No differences in the total neutrophils were detected when comparing COVID-19(−) and recovered COVID-19(+) participants (Figure 4A). On the other hand, the total lymphocytes were lower in the samples from those who had received the second vaccine dose; even in the COVID-19(−) participants, a significant difference was detected compared with the samples without vaccine (*p* = 0.0004) (Figure 4B).

Various inflammatory biomarkers, such as the neutrophil–lymphocyte ratio (NLR), have been investigated as independent predictors for the prognosis of systemic inflammatory diseases [14,15]. When we analyzed the NLR in the participants naïve to COVID-19, we detected a tendency to increase with the first vaccine dose compared with the nonvaccinated samples, and this tendency was maintained following the second vaccine dose. Meanwhile, in the recovered COVID-19(+) participants, the NLR decreased with the first vaccine dose; however, following the second vaccine dose, it returned to a value similar to that in the nonvaccinated samples (Figure 4C).

### 3.3. Effector-Memory B-Lymphocytes Increase with First Vaccine Dose, but There Are No Differences with Second Vaccine Dose

Vaccines aim to generate protective antibodies in the long term; in this sense, the participation of B-lymphocytes is important. Therefore, we analyzed the proportions of total B-lymphocytes (CD19+), as well as memory B-lymphocytes (CD19+CD21+CD27+) and effector-memory B-lymphocytes (CD19+CD21−CD27+). A similar proportion of CD19+ B-lymphocytes was detected in the group of recovered COVID-19(+) participants. However, a lower percentage of CD19+ B-lymphocytes was detected in the samples with the second vaccine dose from the participants with COVID-19(−) compared with that in the nonvaccinated samples (*p* = 0.0189) and the samples with the first vaccine dose (*p* = 0.0097) (Figure 5A).

In the case of memory B-lymphocytes (CD19+CD27+CD21+), a similar proportion of this population was detected between the COVID-19(−) and recovered COVID-19(+) participants. In addition, in the participants with COVID-19(−), a decrease was detected in the samples with the first and second vaccine dose compared with the nonvaccinated samples (*p* = 0.0166 and *p* = 0.0259, respectively). In contrast, the recovered COVID-19(+) participants (Figure 5B) show only a tendency to decrease.

Finally, concerning the effector-memory B-lymphocytes (CD19+CD27+CD21−), a similar proportion was detected when comparing the COVID-19(−) and recovered COVID-19(+) participants in the same group (nonvaccinated, with the first vaccine dose, and with the second vaccine dose) and when comparing the samples from the second dose with the samples from the nonvaccinated. A higher proportion of effector-memory B-lymphocytes was detected in the samples with the first vaccine dose (in both COVID-19(−) and recovered COVID-19(+) participants) compared with that in the nonvaccinated samples (*p* = 0.0214 and *p* = 0.0265, respectively) and compared with the samples with the second vaccine dose (*p* = 0.0362 and *p* = 0.0457, respectively) (Figure 5B). Moreover, we analyzed the samples considering age stratification, and no differences were found (Appendix A).

We analyzed the expression of FcRL4 in both memory B-lymphocytes and effector-memory B-lymphocytes. In participants naïve to COVID-19, we detected a lower percentage of FcRL4+ memory B-lymphocytes in the first and second vaccine doses than the samples without vaccine (*p* = 0.239 and *p* = 0.0171, respectively). On the other hand, in participants with previous COVID-19, only a tendency to decrease was detected (Figure 5C). In the case of FcRL4+ B-lymphocytes with an effector-memory phenotype, a significant increase in this population was detected following the first and second vaccine doses when compared with the nonvaccinated sample from recovered COVID-19(+) participants (*p* < 0.0001 and *p* = 0.0037, respectively), and an increase in the second vaccine dose was detected when compared with the nonvaccinated samples from the COVID-19(−) participants (*p* = 0.035) (Figure 5C).

### 3.4. T-Lymphocytes and Follicular Helper T-Lymphocytes (TFH) Increase with First Dose of Vaccine against SARS-CoV-2 in COVID-19(−) Participants, and TFH Decreases with Second Dose

To effectively activate B-lymphocytes, interaction with T-lymphocytes, particularly the follicular helper T-lymphocytes (TFH), is required. We analyzed the proportions of T-lymphocytes, TFH, and memory T-lymphocytes.

When comparing the proportion of T-lymphocytes between the COVID-19(−) and recovered COVID-19(+) participants, no differences were detected in the variously vaccinated groups. However, an increase in this population was detected in the samples with the first and second vaccine doses compared with the samples from the nonvaccinated COVID-19(−) participants (*p* = 0.003 and *p* = 0.0246, respectively). Additionally, an increase in T-lymphocytes was detected in the samples with the second dose compared with the samples from the nonvaccinated recovered COVID-19(+) participants (*p* = 0.0168) (Figure 6A).

Concerning TFH, in the participants with a history of COVID-19, a similar proportion between the samples with the first vaccine dose and the samples from the nonvaccinated participants was detected, as well as a tendency to decrease the TFH from the recovered COVID-19(+) patients in the samples with the second vaccine dose compared with the samples from the nonvaccinated participants and the samples from the first-dose vaccinated participants (Figure 6B). However, in the participants naïve to COVID-19, we detected a tendency for the proportion of TFH to increase in the first-dose vaccinated samples relative to the nonvaccinated samples. In contrast, in the samples with the second vaccine dose, a significant decrease was detected when comparing the samples with the first vaccine dose and the samples from the nonvaccinated participants (*p* < 0.0001 and *p* = 0.0018, respectively) (Figure 6B). After adjustment by background of COVID-19 and effector-memory B lymphocytes, per unit of change on THF, we observed a significant decrement of 0.0357 (IC95%-0.0636; −0.0077, *p* = 0.012) on the OD. The average difference in the OD among those with and without COVID-19 background was 0.6347 (IC95% 0.3861; 0.8833, *p* < 0.001). No difference on the OD by effector-memory B-lymphocytes was noted. Finally, when analyzing the proportions of memory T-lymphocytes in both COVID-19(−) and recovered COVID-19(+) participants, no differences were found between the vaccinated and nonvaccinated samples (Figure 6C). Furthermore, we analyzed the samples considering age stratification, and no differences were found (Appendix A).

### 3.5. Changes in Parameters by Longitudinal Analysis throughout Follow-Up of Vaccination Schedule

To analyze the immunological changes induced by the vaccine throughout the vaccination schedule, we performed a longitudinal analysis. We considered 14 participants with three samples (pre-vaccine sample and following the first and second vaccine doses) and 36 participants with two samples (10 with pre-vaccine and first vaccine dose samples, and 26 with first and second vaccine dose samples). Concerning the OD, when analyzing the participants naïve to COVID-19, a significant increase in the OD was detected with both vaccine doses (*p* < 0.0001). In contrast, in the recovered COVID-19(+) participants, this increase was only significant with the first vaccine dose (*p* = 0.0039), although there was an increase in the second dose (Figure 7).

Regarding the total count of neutrophils and lymphocytes, no differences were detected throughout the follow-up during the vaccination scheme (Figure 8A,B). Meanwhile, in the NLR, a tendency to increase was detected in the participants with COVID-19(−) with the first and second vaccine doses. Conversely, in the participants who recovered from COVID-19(+), a decrease with the first vaccine dose was detected (*p* = 0.0312), and with the second vaccine dose, the proportion was similar to pre-vaccinated subjects (Figure 8C).

When analyzing the B-lymphocytes, a significant decrease was detected throughout the vaccination scheme in both COVID-19(−) (first vaccine dose, *p* = 0.0049; and second vaccine dose, *p* < 0.0001) and recovered COVID-19(+) (first vaccine dose, *p* = 0.0156; and second vaccine dose, *p* = 0.014) participants (Figure 9A). Moreover, in COVID-19(−) participants, the memory of B-lymphocytes was decreased in the first vaccine dose (*p* = 0.0005), and no differences were detected with the second vaccine dose (Figure 9B). In addition, in the effector-memory B-lymphocyte subpopulation, in both the COVID-19(−) and recovered COVID-19(+) participants, we detected a tendency to increase with the first vaccine dose, whereas, with the second vaccine dose, the percentages decrease to values similar to pre-vaccine proportions (Figure 9B).

When analyzing memory B-lymphocytes, no changes in the FcRL4 expression were detected in the groups studied (Figure 9C). Meanwhile, the proportion of effector-memory B-lymphocytes FcRL4+ in the COVID-19(−) significantly increased with the first (*p* = 0.0122) and second vaccine doses (*p* = 0.0029). In contrast, in the recovered COVID-19(+) participants, only a tendency to increase with the vaccine doses was detected (Figure 9C).

Finally, when analyzing the proportion of T-lymphocytes in COVID-19(−) participants, a significant increase was detected with the first vaccine dose (*p* = 0.0046), whereas in the recovered COVID-19(+) participants, a significant increase was detected with the first and second vaccine doses (*p* = 0.0078, and *p* = 0.0419, respectively) (Figure 10A). In the case of TFH, in the participants naïve to COVID-19, a tendency to increase with the first vaccine dose was detected, followed by a decrease with the second vaccine dose (*p* < 0.0001), whereas in the recovered COVID-19(+) participants, a tendency to decrease was detected with both the first and second vaccine doses (Figure 10B). In COVID-19(−) participants, we detected a tendency for memory T-lymphocytes to increase following the first vaccine dose, followed by a significant decrease with the second vaccine dose (*p* = 0.0284), whereas in the recovered COVID-19(+) participants, a tendency to increase with both the first and second vaccine doses was detected (Figure 10C).

## 4. Discussion

In recent years, the SARS-CoV-2 pandemic has had a meaningful impact on global health. Severe COVID-19 is characterized by acute respiratory distress syndrome, multi-organic failure, and even death [1,16], whereas long-term sequelae have been reported in the case of moderate disease [16]. This pandemic continues because of the high transmissibility, emergence of new variants, and presence of asymptomatic carriers. Therefore, vaccination is the most viable strategy to prevent or diminish the disease. In the present study, we analyzed how different vaccine platforms induce short- and medium-term changes in the B- and T-lymphocyte populations, which might help explain effective immune response postvaccination.

Progress in vaccine generation has resulted in several types of SARS-CoV-2 vaccines approved for emergency use [17,18]. Our study analyzed the immune response induced by Pfizer BNT162b2, which uses mRNA technology and a lipid nanoparticle delivery platform, and the Astra Zeneca AZD1222 and Gam-COVID-Vac (Sputnik V) vaccines that contain viral DNA within nonreplicant adenovirus vectors. They all encode the production of SARS-CoV-2 spike protein, which is the main target for neutralizing antibodies.

The development of adaptive immune response, encompassing neutralizing antibodies and B- and T-lymphocytes, is crucial to controlling and clearing viral infections. It has been reported that individuals infected with SARS-CoV-2 develop an antibody titer around 15 days after the symptoms onset [3]. It is well-known that the antibody titer diminishes over time [2,19]; as expected, the participants included in our study with a history of COVID-19 infection showed the presence of IgGs related to SARS-CoV-2 before vaccination. When analyzing the samples at the first vaccine dose, participants with previous SARS-CoV-2 infection might develop a more rapid and sustained response to a COVID-19 vaccine than individuals naïve to COVID-19 since they had increased antibodies in the serum. At the time of administration of the second vaccine dose, all participants naïve to COVID-19 showed a further increase in IgG related to SARS-CoV-2 at similar levels to those observed with the first dose in participants with primary infection. In contrast, in participants with a history of COVID-19, the OD was similar between vaccine doses, suggesting that both groups reached a plateau. Because the neutralizing capacity of the vaccine-induced antibodies has already been well-demonstrated [20], here, we determined the seroresponse to establish IgG production against SARS-CoV-2.

Several meta-analyses suggest that in COVID-19 patients, CD4+ and CD8+ absolute counts may be valuable biomarkers in the prognosis of disease severity and recovery [14,21,22]. Disease severity has been linked to the lymphocyte-to-neutrophil cell count ratio (NLR) [14,23]. In Mexico, Montiel-Cervantes et al. studied the NRL in patients with severe COVID-19 and compared it with those of healthy controls and convalescent patients. The authors concluded that the NLR is a useful biomarker of survival in severe COVID-19 [24]. The NLR parameter depends on the total count of neutrophils and lymphocytes, and when one of these increases or decreases, the NLR changes. In our study, the NLR increased within their normal range at the second vaccine dose in participants naïve to COVID-19 and participants with previous SARS-CoV-2 infection. A slightly lower count of lymphocytes, without clinical significance, could explain the change in the samples taken after the second vaccine dose. It is well-known that the reduction of immune cell counts in the peripheral blood during viral infection may be caused by virus-induced destruction of T-cells or the mobilization of immune cells to the sites of infection, such as the lungs or lymphatic nodules [25]. Thus, a similar phenomenon may happen in a controlled way in the case of immunization.

Immune responses mediated by CD4+ T-lymphocytes have been associated with controlling SARS-CoV-2 infection [19,22]. Here, we analyzed the changes in the proportions of CD4+ T-lymphocytes and the memory phenotype induced by vaccination against COVID-19 in participants naïve to COVID-19 and participants with a history of SARS-CoV-2 infection. Before vaccination, there were no differences in the percentages of CD4+ T-lymphocytes and CD45RO+ memory T-lymphocytes between participants naïve to COVID-19 and participants with a history of SARS-CoV-2 infection. After follow-up, throughout the vaccine schedule, both participants naïve to COVID-19 and participants with previous SARS-CoV-2 infection showed an increase in CD4+ T-lymphocytes following the first and second vaccine doses and maintained the percentage of this cell population, in contrast to that reported in infected participants who showed a decrease in this cell population [25]. Interestingly, both groups tended to increase the percentage of memory CD45RO+CD4+ T-lymphocytes. Memory T-lymphocytes can be divided into central memory and effector-memory types. Effector-memory T-lymphocytes patrol the peripheral blood and tissues, whereas central memory T-lymphocytes target the lymphoid nodules [11,26]. These memory T-lymphocyte subsets live longer than effector T-lymphocytes, and access to secondary lymphoid tissues contributes to recall responses upon booster vaccination or future infection [26,27]. These results could be evidence of the generation of a robust immune response mediated by T-lymphocytes induced by the distinct vaccine platforms that may have implications for long-term protective immunity.

As mentioned above, virus-specific CD4+ T-lymphocytes may differentiate into multiple distinct phenotypes in response to SARS-CoV-2 infection. These include TFH as a critical component of the potent humoral immune response [28]. Interestingly, our study observed an increased frequency of TFH in participants naïve to COVID-19 after the first vaccine dose, possibly because the interaction with T-lymphocytes is required to develop efficient activation of B-lymphocytes. In contrast, this phenomenon was not wholly seen in the participants with a previous history of SARS-CoV-2 infection, possibly because they had memory B cells that could be activated by vaccination. Our results show the frequency of circulating TFH specific for SARS-CoV-2 generated during acute SARS-CoV-2 infection, which has been associated with reduced disease severity [29]. In contrast, after the second vaccine dose, a decrease in TFH was detected in both groups. In line with our findings, it has been reported that during COVID-19 infection, SARS-CoV-2-specific TFH decreases the CCR6 expression favoring homing to the mucosal airway [29], thus reducing their frequency in blood. One limitation of this study was not performing IL-21 analysis in serum samples; it has been shown that IL-21 promotes B and plasma cell development [30], as well as stimulates the differentiation of TFH. It is necessary to consider analyzing the IL-21 in serum samples in an intracellular manner after in vitro stimuli.

Although antibodies are a central component of vaccine efficacy, B-lymphocytes are fundamental to the adaptive immune system. In particular, the memory B-lymphocytes are studied as evidence of long-term immune protective immunization [31]. Our study analyzed the proportion of memory B-lymphocytes throughout the vaccine schedule; we detected no differences between participants naïve to COVID-19 and participants with a history of SARS-CoV-2 infection. However, both groups show a decrease in CD27+CD21+ B-lymphocytes after vaccination. The peripheral decrease in CD27+CD21+ B-lymphocytes could be explained by homing to lymphatic nodes to establish germinal centers [32].

After the first vaccine dose, the CD27+CD21− B-lymphocytes, considered effector-memory B-lymphocytes, significantly increased after the first vaccine dose, even in participants with a history of SARS-CoV-2 infection. The effector-memory B-lymphocytes are cells that can quickly and efficiently respond to an antigen recall and differentiate into plasmatic cells [32]; this population is increased due to immunization. However, after the second vaccine dose, the percentage of effector memory was similar to the percentage before immunization. This can be explained because, with the application of the second vaccine dose, part of the effector-memory B-lymphocytes differentiates to plasmatic cells.

We also analyzed the human Fc receptor-like 4 (FcRL4), a molecule member of the family homologous to FcγRI in one or more domains and whose expression is associated with memory B-lymphocytes [33]. Typically, FcRL4 is expressed in memory B-lymphocytes localized in the sub-epithelial regions of lymphoid tissues and tonsils [33]. In our study, we detected similar levels of the expression of FcRL4 in central memory B-lymphocytes. Meanwhile, in the case of effector-memory B-lymphocytes, the expression of FcRL4 increased throughout the vaccination schedule in both groups. Increased FcRL4+ B-lymphocytes have been reported in VIH [34] and rheumatoid arthritis [35,36]. Additionally, these B-lymphocytes are associated with an increase in pro-inflammatory cytokines, such as IL-6 [34]. It is important to point out that the function of FcRL4 has been associated with an enhanced response to the TLR9 and may function to switch B-lymphocyte responsiveness from adaptive BCR-mediated signaling to innate TLR-receptor signaling [33,34]. More studies are needed to determine if this signaling pathway is important in vaccines that use RNA or DNA.

Finally, because immune-bridging studies are currently based only on IgG evaluation, our findings can be used as additional evidence to demonstrate vaccine protective effects. After the second dose, the nonvaccinated subjects showed the same immunological changes observed in recovered COVID-19 patients after the first dose. Further studies analyzing the specificity against RBD inside the cell subsets studied here can be helpful to determine a minimal range of cells (TFH or FcRL4 B lymphocytes) needed to maintain an efficient memory immune response in response to antigen. Despite IgG evaluation helping to demonstrate vaccine efficacy, it is well-known that the levels will decrease over time. In contrast, assessing specific memory cells is vital to predicting the ability to respond to future infections [37,38]. Advances in this area are necessary in cases such as the development of vaccine schedules or booster dose schemes by regulatory authorities, and vaccination based on prioritization subjects such as healthcare staff, army, or vulnerable subjects; or based on public health when vaccine production and supply are insufficient.

## 5. Conclusions

In conclusion, immunization against SARS-CoV-2—independently of the vaccine platform—induces IgG production after the first vaccine dose. According to our data, the specific antibody production could result from the TFH–B-lymphocyte interaction. Following a second vaccine dose, both anti-SARS-CoV-2 antibodies and memory cells (T- and B-lymphocytes) were found in the systemic circulation, possibly to prevent subsequent infections (Figure 11). In addition, the late-stage differentiation of effector-memory B-lymphocytes to plasmatic cells was observed with the expression of FcRL4. Finally, because we did not observe differences between vaccines, the results obtained in this work can be used to expand the number of immunological studies to have more bases to perform immuno-bridging analysis during the evaluation and authorization of vaccine platforms.

## Figures and Tables

**Figure 1 vaccines-10-01761-f001:**
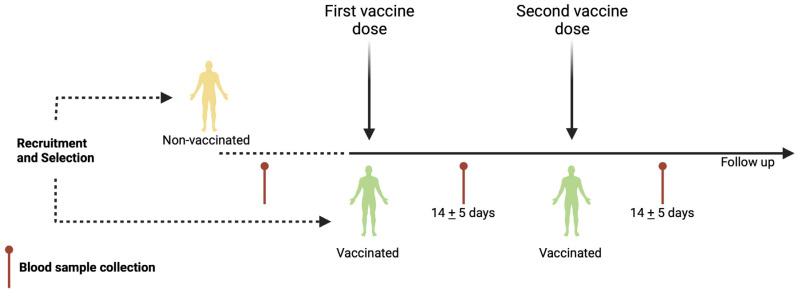
Schematic representation of vaccination schedule. Recruitment, selection, and follow-up of volunteers throughout vaccination schedule. First and second vaccine doses are indicated. Additionally, blood sample collection is shown for nonvaccinated and vaccinated participants. Created using BioRender.com.

**Figure 2 vaccines-10-01761-f002:**
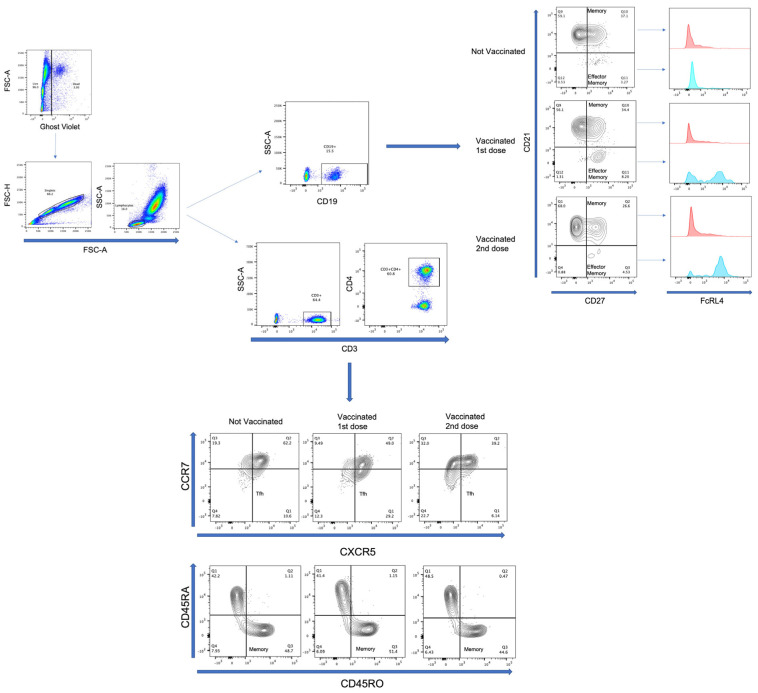
Representative flow cytometry analysis. Cells were gated in an FSC-A vs. Ghost Violet to exclude dead cells; later, in an FSC-A vs. FSC-H dot plot, we excluded doublets. Lymphocyte region was selected using an FSC vs. SSC dot plot. SSC-A vs. CD19 dot plot or SSC-A vs. CD3 followed by CD3 vs. CD4 was made from lymphocyte region. From gated CD19 region, CD27 vs. CD21 plot was made to identify memory (CD21+CD27+) and effector-memory (CD21−CD27+) B-lymphocytes. In addition, expression of FcRL4 was evaluated with histogram from memory and effector-memory B-lymphocytes. On the other hand, from gated CD3, CD4 region, a CXCR5 vs. CCR7 and CD45RO vs. CD45RA plot was made to identify TFH (CCR7−CXCR5+) and memory (CD45RA−CD45RO+) T-lymphocytes, respectively. Analysis of samples from nonvaccinated, vaccinated with the first dose, and vaccinated with the second vaccine dose is shown.

**Figure 3 vaccines-10-01761-f003:**
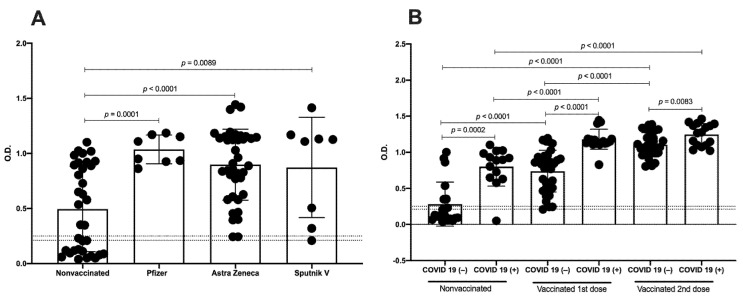
IgGs related to SARS-CoV-2 against RBD domain of S protein induced by distinct vaccine platforms. (**A)** Analysis by OD of samples from nonvaccinated (*n* = 34), vaccinated with Pfizer (*n* = 8), vaccinated with Astra Zeneca (*n* = 38), and vaccinated with Sputnik V (*n* = 8). (**B**) Analysis by OD of samples of nonvaccinated, vaccinated with first vaccine dose, and vaccinated with second vaccine dose from COVID-19(−) and recovered COVID-19(+)participants. Cutoff values are indicated by lines. Mean ± standard deviation of the mean (SD) is shown. Central trend values and dispersion values are indicated in Appendix A.

**Figure 4 vaccines-10-01761-f004:**
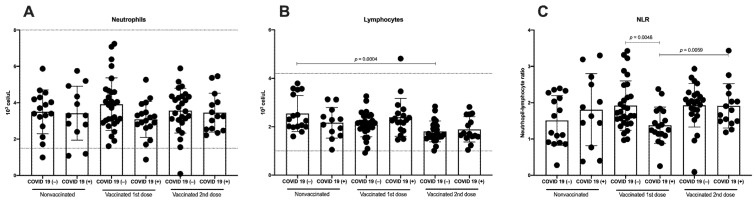
Total count of neutrophils, lymphocytes, and NLR. Analysis of (**A**) total count of neutrophils, (**B**) total count of lymphocytes, and (**C**) and neutrophil-to-lymphocyte ratio (NLR) in samples of nonvaccinated, vaccinated with first vaccine dose, and vaccinated with second vaccine dose from COVID-19(−) and recovered COVID-19(+) participants. Reference values are indicated by lines in (**A**,**B**). Mean ± standard deviation (SD) is shown. Central trend values and dispersion values are indicated in Appendix A.

**Figure 5 vaccines-10-01761-f005:**
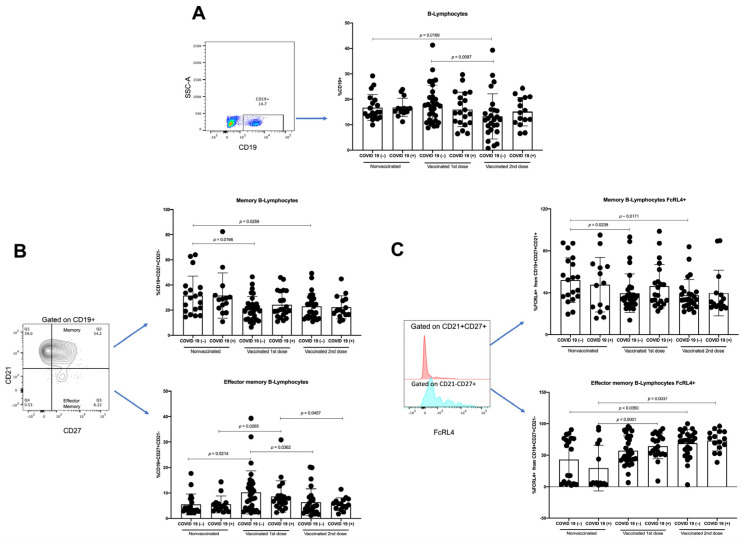
Analysis of B-lymphocytes. As of samples of nonvaccinated, vaccinated with first vaccine dose, and vaccinated with second vaccine dose from COVID-19(−) and recovered COVID-19(+) participants, analysis of (**A**) percentages of B-lymphocytes, obtained from SSC-A vs. CD19 plot, (**B**) percentages of memory B-lymphocytes, and percentages of effector-memory B-lymphocytes, obtained from CD21 vs. CD27 plot gated on CD19+, was performed. Moreover, (**C**) percentages of FcRL4 from memory B-lymphocyte and percentages of FcRL4 from effector-memory B-lymphocytes, obtained from FcRL4 histogram gated on CD21+CD27+ and CD21−CD27+ respectively, was carried out. Mean ± standard deviation (SD) is shown. Central trend values and dispersion values are indicated in Appendix A.

**Figure 6 vaccines-10-01761-f006:**
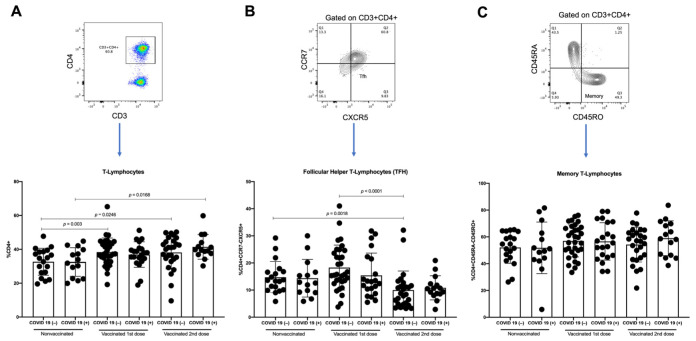
Analysis of T-lymphocytes. As of samples of nonvaccinated, vaccinated with first vaccine dose, and vaccinated with second vaccine dose from COVID-19(−) and COVID-19(+)participants, we analyzed (**A**) percentages of T-lymphocytes obtained from CD4 vs. CD3 plot, (**B**) percentages of follicular helper T-lymphocytes (TFH), obtained from CCR7 vs. CXCR5 plot gated on CD3+CD4+, and (**C**) percentages of memory T-lymphocytes obtained from CD45RA vs. CD45RO plot gated on CD3+CD4+. Mean ± standard deviation (SD) is shown. Central trend values and dispersion values are indicated in Appendix A.

**Figure 7 vaccines-10-01761-f007:**
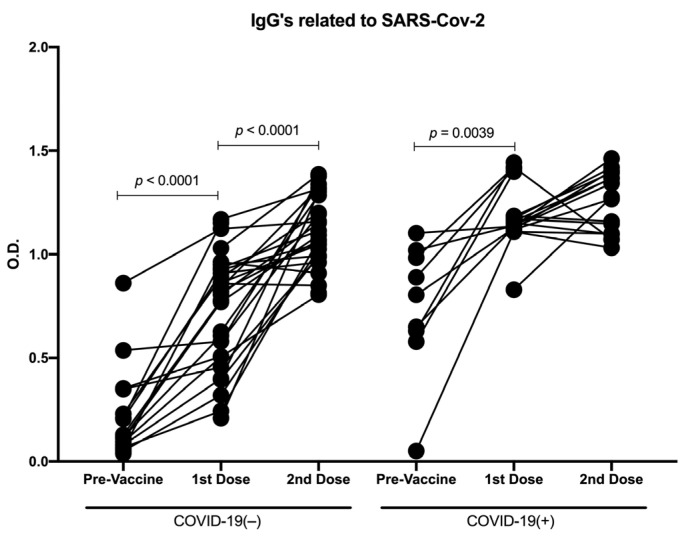
Longitudinal analysis of IgGs related to SARS-CoV-2 against RBD domain of S protein throughout the follow-up of vaccination schedule. Longitudinal analysis by OD of paired samples throughout vaccination scheme (pre-vaccine, first vaccine dose, and second vaccine dose) in COVID-19(−) and recovered COVID-19(+) participants. Central trend values and dispersion values are indicated in Appendix A.

**Figure 8 vaccines-10-01761-f008:**
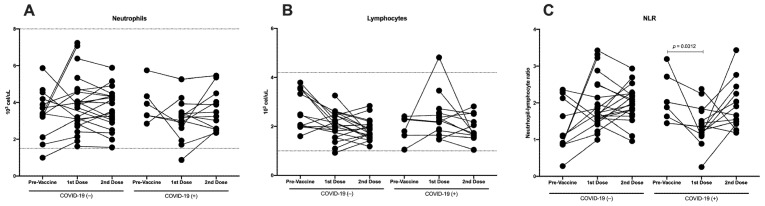
Longitudinal analysis of neutrophils, lymphocytes, and NLR throughout follow-up vaccination schedule. Longitudinal analysis of (**A**) total count of neutrophils, (**B**) total count of lymphocytes, and (**C**) neutrophil-to-lymphocyte ratio (NLR) in paired samples throughout vaccination scheme (pre-vaccine, first vaccine dose, and second vaccine dose) in COVID-19(−) and recovered COVID-19(+) participants. Reference values are indicated by lines in (**A**,**B**). Central trend values and dispersion values are indicated in Appendix A.

**Figure 9 vaccines-10-01761-f009:**
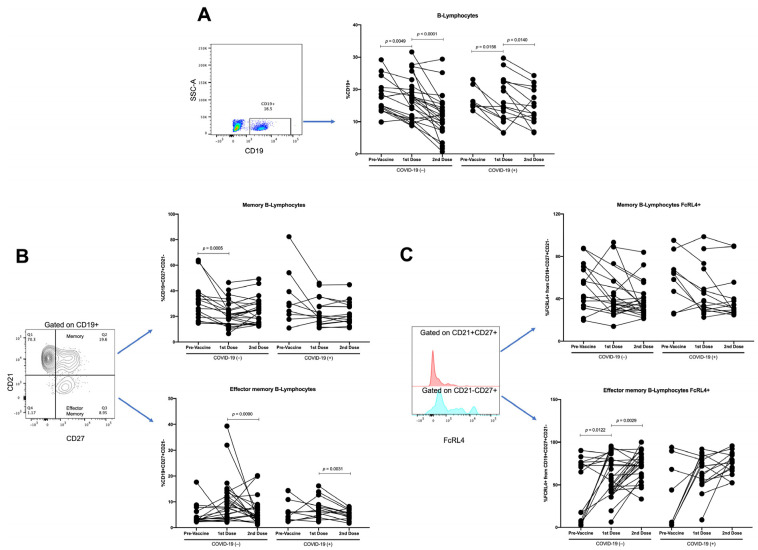
Longitudinal analysis of B-lymphocytes throughout follow-up vaccination schedule. As of paired samples throughout vaccination scheme (pre-vaccine, first vaccine dose, and second vaccine dose) in COVID-19(−) and recovered COVID-19(+) participants, we analyzed (**A**) percentages of B-lymphocytes, obtained from SSC-A vs. CD19 plot, and (**B**) percentages of memory B-lymphocytes and percentages of effector-memory B-lymphocytes, obtained from CD21 vs. CD27 plot gated on CD19+. Moreover, (**C**) percentages of FcRL4 from memory B-lymphocyte, and percentages of FcRL4 from effector-memory B-lymphocytes, obtained from FcRL4 histogram gated on CD21+CD27+ and CD21−CD27+, respectively, was carried out. Central trend values and dispersion values are indicated in Appendix A.

**Figure 10 vaccines-10-01761-f010:**
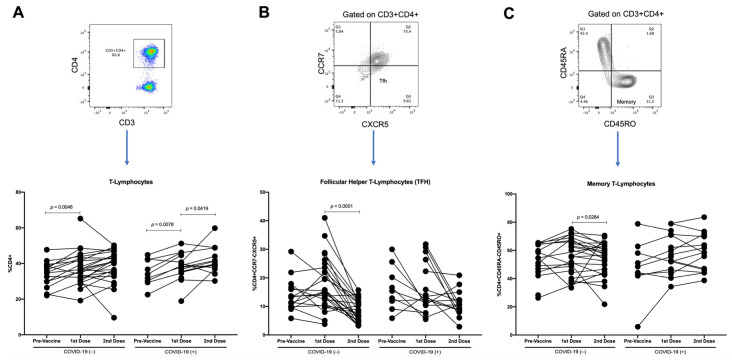
Longitudinal analysis of T-lymphocytes throughout follow-up vaccination schedule. As of paired samples throughout vaccination scheme (pre-vaccine, first vaccine dose, and second vaccine dose) in COVID-19(−) and recovered COVID-19(+) participants, we analyzed (**A**) percentages of T-lymphocytes obtained from CD4 vs. CD3 plot, (**B**) percentages of follicular helper T-lymphocytes (TFH), obtained from CCR7 vs. CXCR5 plot gated on CD3+CD4+, and (**C**) percentages of memory T-lymphocytes obtained from CD45RA vs. CD45RO plot gated on CD3+CD4+. Central trend values and dispersion values are indicated in Appendix A.

**Figure 11 vaccines-10-01761-f011:**
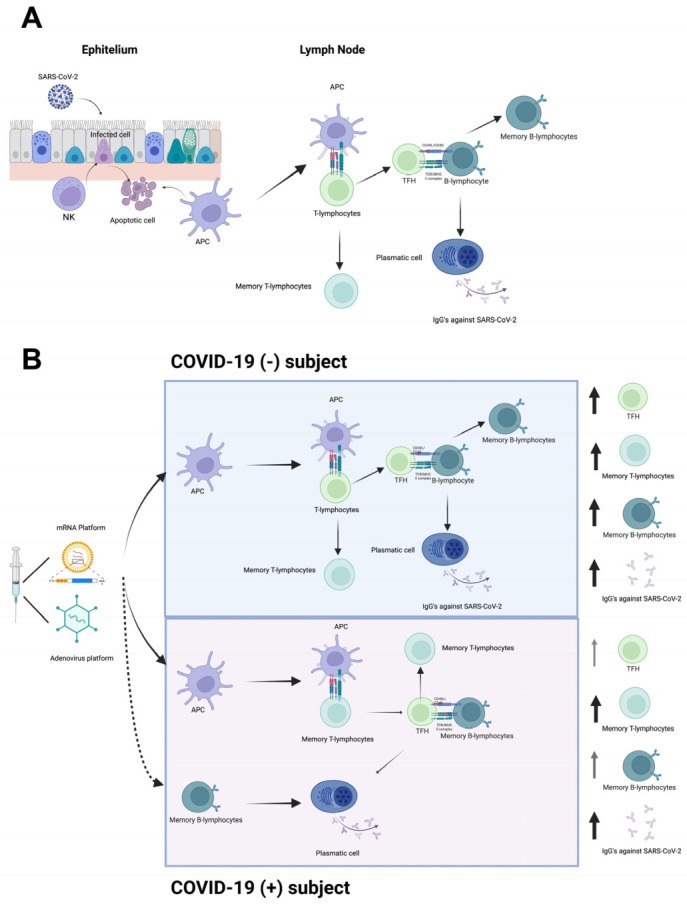
Proposal model of T- and B-lymphocytes as central players after SARS-CoV-2 vaccination. (**A**) Immune response against SARS-CoV-2. Infected cells might be recognized and eliminated by NK cells, followed by recruitment of DCs. DCs become APCs and migrate to lymphatic node to present viral antigens to T-lymphocytes and generate an adaptive immune response composed by CD4+ T-lymphocytes. TFH interacts with B-lymphocytes to produce antibodies against the virus. (**B**) Immune response induced by vaccination. Vaccine induces production of high levels of S protein, and adjuvants enhance recruitment and differentiation of DCs into APCs. In COVID-19(−) subjects, APCs present the antigen to T-lymphocytes. TFH helps S protein-specific B-lymphocytes to differentiate into plasmatic cells and promote production of IgGs against S protein. In contrast, in recovered COVID-19(+) subjects, SARS-CoV-2 specific memory T-lymphocytes, and memory B-lymphocytes developed after infection, could be activated by vaccine application and quickly and efficiently respond to antigen recall. NK, natural killer cell; DC, dendritic cell; APC, antigen-presenting cell; TFH, follicular helper T-lymphocyte. Created using BioRender.com.

**Table 1 vaccines-10-01761-t001:** Demographic data of vaccinated and nonvaccinated groups.

	Not Vaccinated *n* = 34	Vaccinated 1st Dose *n* = 54	Vaccinated 2nd Dose *n* = 43
Age (years) Mean + range	35 (20–49)	37 (20–62)	39 (23–70)
Men	14 (41%)	18 (33%)	15 (34%)
Age (years) Mean + range	35 (20–49)	36 (20–51)	36 (24–51)
Women	20 (59%)	36 (67%)	28 (66%)
Age (years) Mean + range	35 (20–49)	37 (23–62)	39 (23–60)
COVID-19 positive	14 (41%)	21 (38%)	15 (36%)
COVID-19 negative	20 (59%)	33 (62%)	28 (64%)
Pfizer-BioNTech (BNT162b2)	-	8 (15%)	11 (25%)
Astra Zeneca (AZD1222 Covishield)	-	38 (70%)	27 (63%)
Sputnik V (Gam-COVID-Vac)	-	8 (15%)	5 (12%)

## Data Availability

Not applicable.

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
