# Peer review of "Effector-Memory B-Lymphocytes and Follicular Helper T-Lymphocytes as Central Players in the Immune Response in Vaccinated and Nonvaccinated Populations against SARS-CoV-2"

_vaccines, 2022, doi:10.3390/vaccines10101761_

Round 1
Reviewer 1 Report
The authors assessed the central role of effector memory B lymphocytes and follicular helper T lymphocytes in immune responses in vaccinated and unvaccinated populations against SARS-CoV-2. They collected blood samples from donors at 3-time points post-vaccinations with the SARS-CoV-2 vaccine (BNT162b2; AZD1222 Covishield; Gam-COVID-Vac). The authors concluded that after the first vaccine dose, SARS-CoV-2 immunization induced IgG production, and this could be mediated by Tfh and effector memory B-lymphocytes. To further validate this conclusion, new data and analysis need to be added.
1. The age span of donors is relatively large, and it is necessary to analyze the results of different age stratification, such as< 30y and >30y.
2. The representative flow charts are required for each Figure.
3. For the recovered COVID-19 patients mentioned in the article, the specific recovery time should be listed, rather than just the severity of symptoms.
4. Tfh cells can produce large amount of IL-21, which promotes the proliferation and differentiation of B cells into plasma cells. The authors should detect serum IL-21 levels in six different groups.
5. The layout of the figures in the full text is not suitable for publication. Figures and figures legends should be highly revised, to make the figures more readable and the legends more detailed.
Reviewer 2 Report
Presented manuscript explored the human immune responses towards three distinct anti-SARS-CoV-2 vaccine platforms through analyzing IgG production, total neutrophil/lymphocyte quantitation, and multi-panel lymphocyte phenotyping on both short- and medium-terms. Authors presented the data nicely throughout the manuscript and the manuscript is well structured. The rationale of this work is impressive and makes the work more original.
Publication of this review manuscript is recommended following minor suggestions:
1. The introduction of the “follow-up” within the manuscript is quite confusing. It is obvious from Figure 1, that it was scheduled post-vaccination following the second dose. Nevertheless, authors presented follow-up data at three levels pre-vaccination, first dose and second-dose vaccination based on selected 76 participants. Authors should clarify more such approach as well as mention the adopted criteria for selecting those follow-up participants including their demographic data.
2. Following the first point, the time period of the follow-up approach (days, week, ….?) should be highlighted by the authors.
3. Intervals between dosing should be also stated within the manuscript.
4. In Table 1, the authors are advised to rationalize the inclusion of single participant being vaccinated by an unidentified vaccine other than the three designated anti-SARS-CoV-2 vaccine platforms.
5. Within the discussion section, authors are advised to compare their findings with reported studies exploring immunization against SARS-CoV-2 close virus members.
6. Authors illustrated that there were no significant differences with the memory lymphocytes in previous COVID-19 participants following the second vaccination dose, how this could guide the third and even the fourth booster doses advised by regulatory bodies?
Reviewer 3 Report
This paper entitled “Effector Memory B-Lymphocytes and Follicular Helper T-Lymphocytes as Central Players in the Immune Response in Vaccinated and Non-Vaccinated Populations Against SARS-CoV-2” consists of a detailed description of the humoral responses of a cohort of 64 volunteers, including both COVID-19 negative and positive patients, when subjected to vaccination against SARS-CoV-2.
Although the title of this paper sounds interesting, the data presented are a way below the expectation. This is a descriptive report that doesn’t fit with the scoop of MDPI Vaccines. It would likely be better received in a more medically oriented journal.
The design and the rationale of the study suffer from a lack of precise objectives and scientific organization of the experiments. The rationale of comparison of few individuals primo-vaccinated (8) or dual-vaccinated (11) with mRNA vaccine with those vaccinated with Adenovirus-based vaccines, is not clear. The composition of the non-vaccinated group (naïve non-infected, infection negative participants mild or asymptomatic Covid convalescents) is not clear. The time between the first and second vaccination and between infection and vaccination are not indicated.
As indicated by the authors in the discussion, it is known that individuals infected with SARS-CoV-2 develop an antibody titer, around 15 days after symptoms onset, and that the antibody titer diminishes over time. In the study presented here, the date of infection of the COVID-19 subjects is unknown, hence the quantities measured IgG reported in this work is somewhat meaningless. It would have been informative to know when the subjects COVID-19 positive were diagnosed. Furthermore, the interval of time between the first and second vaccination is never cited, which would also influence the titers measured.
The statement that “distinct vaccine platforms induced similar production of antibodies against SARS-CoV-2” is unrealistic. In Figure 3A, the authors should differentiate between the negative and positive patients. This would possibly reduce the wide data spread.
However, concerning the IgGs against the RBD domain of S protein of SARS-CoV-2 (Figure 7), it is interesting to note that detected levels in COVID-19(-) subjects after the second vaccination are close to those seen in COVID-19(+) recovered participants after the first dose.
Round 2
Reviewer 1 Report
All the questions answered or added in discussion.